# A New Early Predictor of Fatal Outcome for COVID-19 in an Italian Emergency Department: The Modified Quick-SOFA

**DOI:** 10.3390/microorganisms10040806

**Published:** 2022-04-12

**Authors:** Matteo Guarino, Benedetta Perna, Francesca Remelli, Francesca Cuoghi, Alice Eleonora Cesaro, Michele Domenico Spampinato, Martina Maritati, Carlo Contini, Roberto De Giorgio

**Affiliations:** 1Department of Translational Medicine, St. Anna University Hospital of Ferrara, University of Ferrara, 44121 Ferrara, Italy; grnmtt@unife.it (M.G.); benedetta.perna@unife.it (B.P.); francesca.cuoghi@edu.unife.it (F.C.); aliceeleonora.cesaro@edu.unife.it (A.E.C.); spmmhl@unife.it (M.D.S.); 2Department of Medical Sciences, St. Anna University Hospital of Ferrara, University of Ferrara, 44121 Ferrara, Italy; francesca.remelli@unife.it; 3Infectious and Dermatology Diseases, St. Anna University Hospital of Ferrara, University of Ferrara, 44121 Ferrara, Italy; martina.maritati@unife.it (M.M.); carlo.contini@unife.it (C.C.)

**Keywords:** COVID-19, in-hospital mortality, MqSOFA, nCOV19, NEWS2, SARS-CoV-2

## Abstract

Background: Since 2019, the novel severe acute respiratory syndrome coronavirus 2 (SARS-CoV-2) is causing a rapidly spreading pandemic. The present study aims to compare a modified quick SOFA (MqSOFA) score with the NEWS-2 score to predict in-hospital mortality (IHM), 30-days mortality and recovery setting. Methods: All patients admitted from March to October 2020 to the Emergency Department of St. Anna Hospital, Ferrara, Italy with clinically suspected SARS-CoV-2 infection were retrospectively included in this single-centre study and evaluated with the MqSOFA and NEWS-2 scores. Statistical and logistic regression analyses were applied to our database. Results: A total of 3359 individual records were retrieved. Among them, 2716 patients were excluded because of a negative nasopharyngeal swab and 206 for lacking data; thus, 437 patients were eligible. The data showed that the MqSOFA and NEWS-2 scores equally predicted IHM (*p* < 0.001) and 30-days mortality (*p* < 0.001). Higher incidences of coronary artery disease, congestive heart failure, cerebrovascular accidents, dementia, chronic kidney disease and cancer were found in the deceased vs. survived group. Conclusions: In this study we confirmed that the MqSOFA score was non-inferior to the NEWS-2 score in predicting IHM and 30-days mortality. Furthermore, the MqSOFA score was easier to use than NEWS-2 and is more suitable for emergency settings. Neither the NEWS-2 nor the MqSOFA scores were able to predict the recovery setting.

## 1. Introduction

The new zoonotic coronavirus (SARS-CoV-2) responsible for coronavirus disease (COVID-19) is a strain of coronavirus not previously seen in humans [1,2,3]. Different hypotheses about its origin have been proposed, but the direct ancestral virus has not been identified yet [4,5]. The virus originated in Wuhan, Hubei Province, China, and spread rapidly throughout the world, causing over 452 million global cases with different clinical presentation and 6.03 million deaths, with different mortality rates, in almost every country in the world, including Europe and particularly in Italy [1,2,3,4,5]. Common complications include acute respiratory distress syndrome (ARDS), acute kidney injury, elevated liver enzymes, delirium/encephalopathy, thrombosis, and cardiac injuries.

Several risk factors for COVID-19 severity have been described in the literature. In particular, three risk categories have been proposed: (i) “life-style factors” (e.g., smoking habit and diet-related obesity); (ii) demographic factors (e.g., age, male gender, post-menopausality); (iii) comorbidities (e.g., hypertension, diabetes, coronary artery disease, cerebrovascular disease, chronic kidney disease and chronic obstructive pulmonary disease) [6].

As a patient affected by COVID-19 may rapidly worsen, an early assessment of illness severity is important for risk stratification and decision-making. Several studies have proposed different clinical risk scores (e.g., NEWS-2, SIRS, SOFA and qSOFA) to predict fatal outcomes in patients with COVID-19 [6,7,8,9]. However, there is a lack of evidence supporting their use in patients with SARS-CoV-2 infection. The NEWS-2 is the only score, which seems to predict disease severity and in-hospital-mortality at emergency department admission [7].

A recently proposed tool, i.e., a modified qSOFA (MqSOFA) score, added the SpO_2_/FiO_2_ ratio to the usual qSOFA parameters (systolic blood pressure ≤ 100 mmHg, respiratory rate ≥ 22 and acute altered mentation), is superior to qSOFA and easier to use than the NEWS-2 score in assessing the risk of in hospital mortality (IHM) in septic patients [10,11]. Furthermore, the SpO_2_/FiO_2_ ratio is considered to be a promising tool for predicting the risk of mechanical ventilation in patients infected with COVID-19 [12].

As sepsis and COVID-19 share many clinical features [13], the primary aim of the present study was to propose the MqSOFA score for the early assessment of COVID-19 patients and compare this tool with the NEWS-2 score to predict the overall risk of IHM and 30-day mortality (see Table 1 for features of the involved tools). Furthermore, as a secondary aim, we analysed the ability of the involved scores in predicting the recovery setting.

## 2. Materials and Methods

All patients admitted from March to October 2020 to the Emergency Department of St. Anna Hospital, Cona, Ferrara, Italy with clinical suspected SARS-CoV-2 infection were retrospectively included in this single-centre study. We retrieved a total number of 3359 individual records. Among them, 2716 patients were excluded because of a negative molecular nasopharyngeal swab and other 206 for incomplete report of vital parameters; thus, 437 patients were eligible for the study. The MqSOFA and NEWS-2 scores were assessed for all the involved patients and a “high-risk” level (i.e., MqSOFA ≥ 2, NEWS-2 ≥ 7) was determined. Patients’ comorbidities were assessed using Charlson Comorbidity Index [14]. As the S/F ratio loses significance in intubated patients (the FiO_2_ parameters is induced by a ventilator), this subgroup was excluded from the analysis.

As a retrospective study, not actively involving patients, this research was reviewed by our ethics review board, which deemed unnecessary the request of individual informed consent.

### Statistical Analysis

Continuous variables were presented using mean and standard deviation, and categorical variables with frequency and percentage. The characteristics of individuals were compared according to IHM and 30-day mortality using the *t*-test and chi-squared test, as appropriate. The comparison between the tested scores (NEWS-2 and MqSOFA) was performed through the assessment of sensitivity, specificity, Positive Predictive Value (PPV), Negative Predictive Value (NPV), diagnostic accuracy, diagnostic Odds Ratio (OR) and Youden Index for both IHM and 30-day mortality. The associations of the two scores with IHM and 30-day mortality were evaluated through Cox regression analysis. The results were presented using Hazard Ratios (HR) and 95% Confidence Interval (95% CI). Through the goodness-of-fit test, the proportionality hazard assumption was demonstrated. The Model 1 was unadjusted, while the Model 2 was adjusted for potential confounders (age, sex, and Charlson Comorbidity Index). In the Kaplan–Meier survival analysis, performed for both scores, the enrolled individuals were stratified into two groups: patients with high score value and patients with low score value. Statistical analyses were conducted with the software R 3.5.0 and a *p*-value < 0.05 was considered statistically significant.

## 3. Results

A total of 437 patients were included in this single-centre, retrospective study. Among included patients, 231 were males (52.9%) and 206 were females (47.1%) with a mean age of 47.8 ± 18.5 years. Patients’ outcomes were described in terms of IHM (93 patients, 21.3%) and 30-days mortality (96 patients, 28.2%) (other features of the sample are described in Table 2).

The following section will be divided in two paragraphs to clarify the different findings (summarized in Table 3) related to the primary outcome (i.e., IHM and 30-day mortality).

### 3.1. Characteristics of Patients in Relation to the Primary Outcome

#### 3.1.1. IHM

In this subset, age was significantly higher in the group of deceased vs. survived patients (62.2 ± 10.7 vs. 43.8 ± 18.3 years, *p* < 0.001). Moreover, deceased patients presented more comorbidities than survived ones (CCI: 3.86 ± 2.42 vs. 2.43 ± 1.87, *p* < 0.001). Among comorbidities, a higher incidence of coronary artery disease (CAD) (17.4% vs. 8.2%, *p* = 0.017), congestive heart failure (CHF) (16.1% vs. 6.5%, *p* = 0.006), cerebrovascular accidents (CVA) (36.6% vs. 16.5%, *p* < 0.001), dementia (43.0% vs. 19.4%, *p* < 0.001), chronic kidney disease (CKD) (20.4% vs. 6.2%, *p* < 0.001) and cancer without metastasis (18.3% vs. 7.0%, *p* = 0.002) was found in the deceased vs. survived subset. NEWS-2 ≥ 7 (44.1% vs. 13.4%, *p* < 0.001) and MqSOFA ≥ 2 (43.3% vs. 12.2%, *p* < 0.001) were more frequent in deceased patients (see Table 3).

#### 3.1.2. 30-Day Mortality

Even in the 30-day mortality subset, age was significantly higher in deceased vs. survived patients (62.3 ± 11.3 vs. 43.7 ± 18.1 years, *p* < 0.001). Deceased patients showed more comorbidities than survived ones (CCI: 3.86 ± 2.26 vs. 2.42 ± 1.91, *p* = 0.009). Among comorbidities, a higher incidence of CAD (17.9% vs. 8.0%, *p* = 0.017), CHF (17.7% vs. 5.9%, *p* = 0.001), CVA (41.7% vs. 14.8%, *p* < 0.001), dementia (45.8% vs. 18.3%, *p* < 0.001), CKD (20.8% vs. 5.9%, *p* < 0.001) and cancer without metastasis (15.6% vs. 7.7%, *p* = 0.032) was detectable in the deceased vs. survived group. NEWS-2 ≥ 7 (42.7% vs. 13.5%, *p* < 0.001) and MqSOFA ≥ 2 (41.7% vs. 12.3%, *p* < 0.001) were more frequent in deceased patients (see Table 3).

### 3.2. Logistic Regression and Supplementary Analysis

The univariate (Model 1) and multivariate (Model 2) logistic regressions were performed for both scores. A NEWS-2 ≥ 2 was independently associated with both IHM (HR 2.37, 95% CI: 1.53–3.66) and 30-day mortality (HR 2.44, 95% CI: 1.60–3.72). Similarly, a MqSOFA ≥ 2 was independently associated with both IHM (HR 2.19, 95% CI: 1.39–3.43) and 30-day mortality (HR 2.23, 95% CI: 1.45–3.44) (Table 4). Data presented in Table 5 (showing the levels of sensitivity, specificity, PPV, NPV, diagnostic accuracy, diagnostic OR and Younden index) highlighted the non-inferiority of MqSOFA vs. NEWS-2 scores in predicting both IHM and 30-days mortality.

According to the mortality risk assessed on the whole sample by Kaplan Meier’s method, the probability of death at 7 days was 6.64%, reaching a 30-day mortality risk of 22.18%. Survival probabilities in overall sample based on NEWS-2 and MqSOFA values were showed in Figure 1 and Figure 2.

Regarding secondary outcome (i.e., recovery setting), neither a high value of NEWS-2 (*p* = 0.135) nor MqSOFA (*p* = 0.960) were significantly associated with the setting of hospitalization.

Assuming an Alpha = 0.5 in the analysed sample of 437 patients, the statistical power of the data analysis was 71.8%.

## 4. Discussion

Since the first cases of pneumonia of unknown origin reported in Wuhan in December 2019, new variants of SARS-CoV-2 have emerged worldwide, requiring sustained attention to their transmissibility and severity [1,2,3,4,5]. To date, there have been 452,201,564 confirmed cases of COVID-19, including 6,029,852 deaths, reported to WHO (World Health Organization 2022) [15]. Currently, the number of new SARS-CoV-2 infections are increasing due to the outbreak of BA1.1 and BA2 Omicron variants that are particularly contagious but less aggressive, because they do not involve the lower airways, than the Delta variant that prevailed during the third wave that had high clinical severity [16,17,18].

Predicting the outcome of SARS-CoV-2 infection is fairly difficult based on current testing and there are multiple factors that come into play. Despite the publication of COVID-19 guidelines in 2020 [19] and the subsequent update in 2021 [20], no validated scores [21,22,23] have been proposed to precisely predict the risk of fatal outcome of patients with SARS-CoV-2 infection in the ED. The main objective of this article was to compare different screening tools (i.e., MqSOFA and NEWS-2) to identify the best performing one. Advantages and limits of MqSOFA have been previously reported [10,11,12,24]. The main limitation of NEWS-2 lies in its complexity. Indeed, although non-invasive, it requires many parameters and scoring ranges [21].

The analysis performed in this study expressed the level of comorbidities by the CCI [14]. Among the considered comorbidities, CAD, CHF, CVA, dementia, CKD and cancer negatively impacted on the prognosis of patients with SARS-CoV-2 infection. As observed by Inciardi et al. patients with underlying CAD showed poorer outcomes compared to those without CAD [25]. Similarly, patients with a previous diagnosis of CHF presented a higher mortality rate, likely due to acute heart failure decompensation [26]. In the literature, correlations between COVID-19 and CVA [27], dementia [28], CKD [29] and cancer [30] have been proposed to date, resulting in a negative predictive factor for patient’s outcome. Additionally, the relationship between COPD and COVID-19 deserves a careful discussion. The results presented in this study showed no significant difference between survived vs. deceased in the sub-cohort of patients affected by COPD, in contrast with data reported in the literature [31,32]. The first reason to explain this discrepancy is that routine medications used in COPD (e.g., inhaled, and systemic corticosteroids, β2-agonists, muscarinic antagonists) may play a protective role even in patients with COVID-19 infection, but additional studies are needed. Secondly, COPD is largely recognized to be a heterogeneous disease with multiple phenotypes (e.g., frequent exacerbators, emphysema-predominant, asthma-COPD overlap). One may argue that each of these phenotypes behaves differently in a COVID-19 infection, but our data lacked information about the spectrum of severity of the disease in these patients. Clearly, we are well aware that the COPD sample size is small (*n* = 56), thus representing a limitation of this finding; further studies are eagerly awaited to define how different phenotypes of COPD could impact on the outcome of COVID-19 patients and whether the chronic medication may represent a protective factor in a SARS-CoV-2 infection.

The multivariate logistic regression indicated that both NEWS-2 ≥ 7 and MqSOFA ≥ 2 were independently negative predictors of a fatal outcome. Indeed, patients with NEWS-2 ≥ 7 have twice the risk of IHM (HR 2.37, 95% CI: 1.53–3.66, *p* < 0.001) and 30-day mortality (HR 2.44, 95% CI: 1.60–3.72, *p* < 0.001). Patients with a MqSOFA ≥ 2 also showed a similar risk of IHM (HR 2.19, 95% CI: 1.39–3.43, *p* < 0.001) and mortality at 30 days (HR 2.23, 95% CI: 1.45–3.44, *p* < 0.001).

Table 5 highlights the levels of sensitivity, specificity, PPV, NPV, diagnostic accuracy, diagnostic OR and Younden index, showing that the two involved scores equally predicted IHM and 30-day mortality. Therefore, it is possible to conclude for a “non-superiority” of NEWS-2 over MqSOFA in terms of short- and medium-term prognosis. A valuable score for ED should be able to detect patients who may rapidly deteriorate and need a higher intensity of care. Thus, tools (i.e., NEWS-2 and MqSOFA) characterized by high sensitivity and PPV are useful in the early management of patients with SARS-CoV-2 infection. Considering the “non-inferiority” of MqSOFA over NEWS-2 and its easy use, this tool was shown to be the most suitable in the emergency setting among the assessed scores.

In our sample, the IHM was extremely variable, ranging from 0 to 194 days; thus, in order to express the overall survival probabilities, we chose to consider two ranges of time (7- and 30-days mortality). Indeed, Figure 1 showed that at 7 days the probability of death in patients with NEWS2 ≥ 7 was more than four times greater than in patients with NEWS-2 < 7 (7 days Risk Ratio = 4.31); at 30 days the probability of death in patients with NEWS-2 ≥ 7 is more than two times greater than in patients with NEWS-2 < 7 (30 days Risk Ratio = 2.80) (*p* < 0.0001). Similarly, Figure 2 showed that at 7 days the probability of death in patients with MqSOFA ≥ 2 is five times greater than in patients with MqSOFA < 2 (7 days Risk Ratio = 5.27); at 30 days the probability of death in patients with MqSOFA ≥ 2 is more than three times greater than in patients with MqSOFA < 2 (30 days Risk Ratio = 3.02) (*p* < 0.0001).

In this analysis, both the involved scores were not able to predict the intensity of care needed by the patient with the SARS-CoV-2 infection. These data contrast with the results of a previous analysis [18]. However, for the sake of clarity, the present dataset is wider than the one presented by Covino et al.; furthermore, we only considered patients directly admitted to ICU (and not after 48-h/7-days) [22].

This research has different strengths: firstly it introduces a simple and totally inexpensive tool in the primary evaluation of COVID-19 patients. This score may be assessed in every setting (in or out-of-hospital) resulting particularly suitable to the ED. Furthermore, the involved cohort has demographic features comparable to the ones presented in literature [1]. We acknowledge some limitations of our study: first, as a retrospective and single-centre analysis, its statistical power is strongly reduced. Second, we excluded a quite high proportion (almost a third) of patients for inadequate data. Third, the S/F ratio has intrinsic limitations related to the SpO_2_ parameter and its high variability in different clinical conditions, e.g., carbon monoxide poisoning or sickle cell anaemia. The SpO_2_ value may also be falsely low in paradoxical pulse, severe anaemia with concomitant hypoxia or in poor finger/nail cleaning. Finally, other conditions that may alter the SpO_2_ parameter include methemoglobinemia, sulfhemoglobinemia, severe hyperbilirubinemia, circulating foetal haemoglobin as well as all the causes of poor peripheral perfusion [10,11,12,22].

## 5. Conclusions

In this single-centre and retrospective study we confirmed that the MqSOFA score was non-inferior to the NEWS-2 score in prediction of both IHM and 30-days mortality. Furthermore, the MqSOFA score was easier to use than the NEWS-2 score making it more suitable for emergency settings. Nevertheless, neither the NEWS-2 nor the MqSOFA scores were able to predict the intensity of care needed by the patient with SARS-CoV-2 infection, so a clinical “case-by-case” evaluation is deemed necessary. Future prospective studies, performed on larger cohorts, are largely awaited to demonstrate the efficacy of a simple and inexpensive score, i.e., MqSOFA, in predicting the outcome of patients with COVID-19.

## Figures and Tables

**Figure 1 microorganisms-10-00806-f001:**
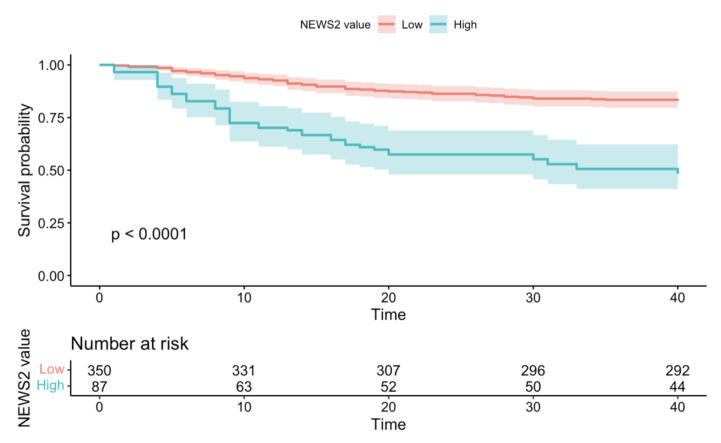
Survival probability in overall sample based on NEWS-2 value.

**Figure 2 microorganisms-10-00806-f002:**
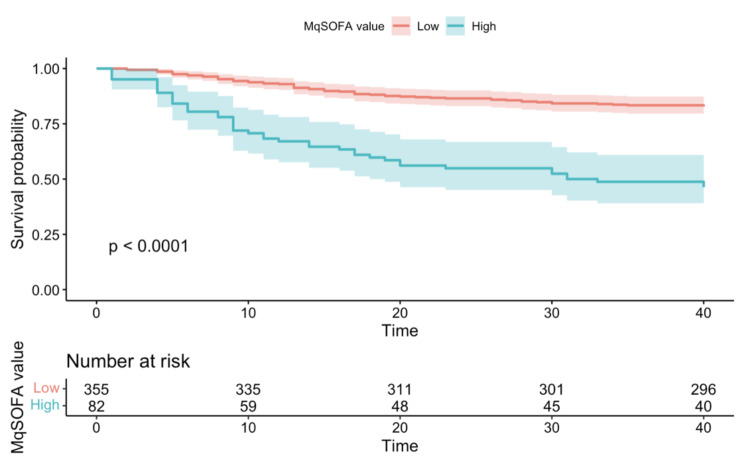
Survival probability in overall sample based on MqSOFA value.

**Table 1 microorganisms-10-00806-t001:** Comparison between MqSOFA and NEWS-2 scores.

MqSOFA							
Parameter						Points	
Blood Pressure ≤ 100 mmHg						1	
Respiratory Rate ≥ 22/min						1	
Altered Mentation						1	
SpO_2_/FiO_2_ ratio			≥316			0	
			236–315			1	
			≤235			2	
**NEWS-2**							
**Parameter**	**3**	**2**	**1**	**0**	**1**	**2**	**3**
Respiratory Rate	≤8		9–11	12–20		21–24	≥25
O_2_ Saturation Scale 1 (%)	≤91	92–93	94–95	≥96			
O_2_ Saturation Scale 2 (%)	≤83	84–85	86–87	88–92 ≥ 93 on air	93–94 on oxygen	95–96 on oxygen	≥97 on oxygen
Supplemental O_2_		Yes		No			
Temperature (°C)	≤35.0		35.1–36.0	36.1–38.0	38.1–39.0	≥39.1	
Systolic Blood Pressure (mmHg)	≤90	91–100	101–110	111–219			≥220
Heart Rate	≤40		41–50	51–90	91–110	111–130	≥131
Level of Consciousness (AVPU)				Alert			Verbal, Pain, Unresponsive

**Table 2 microorganisms-10-00806-t002:** Features of the sample at admission.

Patients, *n*	437
Female, *n* (%)	206 (47.1)
Age (years), mean (SD)	47.79 (18.53)
Hospital Unit, (%)	
*Discharged*	*0 (0)*
*Low-intensity care*	*316 (72.3)*
*Intermediate-intensity care*	*121 (27.7)*
*High-intensity care*	*0 (0)*
Comorbidities	
*CAD (%)*	*44 (10.2)*
*CHF (%)*	*37 (8.5)*
*Peripheral vascular disease (%)*	*11 (2.5)*
*CVA (%)*	*90 (20.8)*
*Dementia (%)*	*106 (24.4)*
*COPD (%)*	*56 (12.8)*
*Connective tissue disease (%)*	*46 (10.6)*
*Peptic ulcer disease (%)*	*5 (1.2)*
*Liver disease (%)*	*7 (1.6)*
*Uncomplicated diabetes mellitus (%)*	*66 (15.2)*
*Diabetes mellitus with end-organ damage (%)*	*25 (5.8)*
*Hemiplegia*	*7 (1.6)*
*Moderate to severe CKD (%)*	*40 (9.2)*
*Cancer without metastasis (%)*	*41 (9.4)*
*Moderate to severe liver disease (%)*	*2 (0.5)*
*Metastatic tumour (%)*	*10 (2.3)*
**CCI, mean (SD)**	2.73 (2.08)
**NEWS-2, mean (SD)**	4.75 (3.11)
**MqSOFA, mean (SD)**	1.76 (1.04)
**LOS, mean (SD)**	14.53 (15.48)

**Table 3 microorganisms-10-00806-t003:** Characteristics of enrolled participants according to IHM and 30-day mortality.

	IHM			30-Days Mortality		
Variables	Survived (*n* = 344)	Deceased (*n* = 93)	*p*	Survived (*n* = 341)	Deceased (*n* = 96)	*p*
Female, *n* (%)	154 (44.8)	52 (55.9)	0.073	152 (44.6)	54 (56.2)	0.056
Age (years), mean (SD)	43.88 (18.26)	62.25 (10.76)	**<0.001**	43.71 (18.14)	62.29 (11.25)	**<0.001**
Comorbidities						
*CAD (%)*	28 (8.2)	16 (17.4)	**0.017**	27 (8.0)	17 (17.9)	**0.009**
*CHF (%)*	22 (6.5)	15 (16.1)	**0.006**	20 (5.9)	17 (17.7)	**0.001**
*Peripheral vascular disease (%)*	7 (2.1)	4 (4.3)	0.398	7 (2.1)	4 (4.2)	0.435
*CVA (%)*	56 (16.5)	34 (36.6)	**<0.001**	50 (14.8)	40 (41.7)	**<0.001**
*Dementia (%)*	66 (19.4)	40 (43.0)	**<0.001**	62 (18.3)	44 (45.8)	**<0.001**
*COPD (%)*	44 (12.8)	12 (12.9)	1.000	45 (13.4)	11 (11.5)	0.752
*Connective tissue disease (%)*	35 (10.3)	11 (11.8)	0.814	33 (9.8)	13 (13.5)	0.388
*Peptic ulcer disease (%)*	4 (1.2)	1 (1.1)	1.000	3 (0.9)	2 (2.1)	0.674
*Liver disease (%)*	6 (1.8)	1 (1.1)	0.995	6 (1.8)	1 (1.0)	0.959
*Uncomplicated diabetes mellitus (%)*	48 (14.1)	18 (19.4)	0.279	46 (13.6)	20 (20.8)	0.117
*Diabetes mellitus with end-organ damage (%)*	18 (5.3)	7 (7.5)	0.571	18 (5.3)	7 (7.3)	0.635
*Hemiplegia*	7 (2.1)	0 (0.0)	0.352	7 (2.1)	0 (0.0)	0.335
*Moderate to severe CKD (%)*	21 (6.2)	19 (20.4)	**<0.001**	20 (5.9)	20 (20.8)	**<0.001**
*Cancer without metastasis (%)*	24 (7.0)	17 (18.3)	**0.002**	26 (7.7)	15 (15.6)	**0.032**
*Moderate to severe liver disease (%)*	0 (0.0)	2 (2.2)	0.065	0 (0.0)	2 (2.1)	0.072
*Metastatic tumour (%)*	6 (1.8)	4 (4.3)	0.290	7 (2.1)	3 (3.1)	0.824
CCI, mean (SD)	2.43 (1.87)	3.86 (2.42)	**<0.001**	2.42 (1.91)	3.86 (2.26)	**<0.001**
NEWS-2 ≥ 7, *n* (%)	46 (13.4)	41 (44.1)	**<0.001**	46 (13.5)	41 (42.7)	**<0.001**
MqSOFA ≥ 2, *n* (%)	42 (12.2)	40 (43.3)	**<0.001**	42 (12.3)	40 (41.7)	**<0.001**

**Table 4 microorganisms-10-00806-t004:** Multivariate logistic regression analysis for the probability of in-hospital and 30-day mortality (NEWS-2 vs. MqSOFA).

	Model 1			Model 2		
	HR	95% CI	*p*	HR	95% CI	*p*
In-hospital mortality						
NEWS-2	3.56	(2.34–5.41)	<0.001	2.37	(1.53–3.66)	<0.001
Age (years)	1.08	(1.06–1.10)	<0.001	1.08	(1.05–1.10)	<0.001
Sex (F)	1.60	(1.06–2.42)	<0.05	0.83	(0.53–1.30)	0.406
CCI	1.22	(1.14–1.31)	<0.001	1.05	(0.96–1.14)	0.273
30-day mortality						
NEWS-2	3.65	(2.42–5.50)	<0.001	2.44	(1.60–3.72)	<0.001
Age (years)	1.08	(1.06–1.10)	<0.001	1.07	(1.05–1.09)	<0.001
Sex (F)	1.58	(1.06–2.38)	<0.05	0.85	(0.55–1.32)	0.472
CCI	1.24	(1.16–1.33)	<0.001	1.08	(0.99–1.17)	0.077
In-hospital mortality						
MqSOFA	3.76	(2.47–5.72)	<0.001	2.19	(1.39–3.43)	<0.001
Age (years)	1.08	(1.06–1.10)	<0.001	1.07	(1.05–1.10)	<0.001
Sex (F)	1.60	(1.06–2.42)	<0.05	0.77	(0.49–1.21)	0.258
CCI	1.22	(1.14–1.31)	<0.001	1.05	(0.97–1.15)	0.233
30-day mortality						
MqSOFA	3.86	(2.56–5.82)	<0.001	2.23	(1.45–3.44)	<0.001
Age (years)	1.08	(1.06–1.10)	<0.001	1.07	(1.05–1.09)	<0.001
Sex (F)	1.58	(1.06–2.38)	<0.05	0.79	(0.51–1.23)	0.292
CCI	1.24	(1.16–1.33)	<0.001	1.08	(1.00–1.18)	0.062

**Table 5 microorganisms-10-00806-t005:** Comparison of NEWS-2 and MqSOFA scores according to in-hospital and 30-day mortality.

	In-Hospital Mortality		30-Day Mortality	
	NEWS-2	MqSOFA	NEWS-2	MqSOFA
Sensitivity	85.1%	85.1%	84.3%	84.2%
Specificity	47.1%	48.8%	47.1%	48.8%
PPV	86.6%	87.8%	86.5%	87.7%
NPV	44.1%	43.0%	42.7%	41.7%
Diagnostic accuracy	77.6%	78.3%	76.9%	77.6%
Diagnostic OR	5.11	5.43	4.78	5.09
Youden index	0.32	0.35	0.31	0.33

## Data Availability

The datasets generated and/or analysed during the current study are not publicly available due to privacy policy but are available from the corresponding author on reasonable request.

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
