# Peer review of "A New Early Predictor of Fatal Outcome for COVID-19 in an Italian Emergency Department: The Modified Quick-SOFA"

_microorganisms, 2022, doi:10.3390/microorganisms10040806_

Round 1

Reviewer 1 Report

This is a very interesting study that could help clinicians in the evaluation of COVID-19 patients using a simple, quick and inexpensive algorithm. The study is well organized and the results from the statistical analysis are clear. Excluding the fact that English should be revised by a native speaker, I have only some minor comments to address to the authors.

The introduction should be more extended and provide more details about risk factors and risk prediction systems and how useful they are in predicting the severity and mortality of the disease.

The parameters evaluated in qSOFA vs MqSOFA should be mentioned.

Lines 32-33: the statement ‘which appears to come from bats’ should be rephrased because despite the discovery of several closely related viruses in bats, the direct evolutionary progenitor of SARS-CoV-2 has not yet been identified.

Line 115-117: I think they are not necessary… at the end of paragraphs 3.1.1 and 3.1.2 the authors should refer to table 3.

Line 141: scores

Author Response

Replies to Reviewers’ comments – “A new early predictor of fatal outcome for COVID-19 in an Italian Emergency Department: the Modified Quick-SOFA” – Microorganisms-1665355 by Guarino et al.

Reviewer #1:

We wish to thank again the Reviewer for her / his insightful comments. The following is a thorough point-by-point reply to each comment.

  1. The introduction should be more extended and provide more details about risk factors and risk prediction systems and how useful they are in predicting the severity and mortality of the disease.

Replay to comment 1: We agree with the Reviewer. Two new sub-paragraphs about risk factors and risk prediction systems have been added to Introduction (see pages 1-2, lines 40-45 and page 2, lines 49-52).

  1. The parameters evaluated in qSOFA vs MqSOFA should be mentioned.

Replay to comment 2: Absolutely agree with the Reviewer. Both qSOFA and MqSOFA parameters have been included in the Introduction (see page 2, lines 53-55).

  1. Lines 32-33: the statement ‘which appears to come from bats’ should be rephrased because despite the discovery of several closely related viruses in bats, the direct evolutionary progenitor of SARS-CoV-2 has not yet been identified.

Replay to comment 3: We agree with this Reviewer’s comment and correct the statement about COVID-19 origin (see Introduction, page 1, lines 32-34).

  1. Line 115-117: I think they are not necessary… at the end of paragraphs 3.1.1 and 3.1.2 the authors should refer to table 3

Replay to comment 4: We thank the Reviewer for the comment. However, we think that a full explanation of these findings highlights the importance of comorbidities in the fatal outcome prediction of patients with COVID-19. Table 3 has been mentioned at the end of both paragraphs 3.1.1 and 3.1.2.

  1. Line 141: scores

Replay to comment 5: All typos have been corrected and a new English proofread has been performed.

Reviewer 2 Report

Guarino and colleagues aimed to evaluate whether mqSOFA is inferior to NEWS2 in predicting the IHM for 30 days under the COVID19 infection. Since the COVID area is an emerging medical problem, all the papers are so important to the field. 

I found the paper nicely written, well-structured, however, some minor grammatical errors have been found throughout the whole draft. 

My main concern about the paper is rather philosophical than scientific and is related to the novelty of the data shown there. I am not sure whether the paper is publishable in its present form - please take into account that similar studies have been done recently, with almost the same results (please refer to Frontiers in Medicine, 2021; JAMA, 2020; JAMA, 2021; Frontiers in Medicine, 2022; Blue Journal/ATS Scholar 2021; NEJM 2020 and much more...) on greater and more homogeneous groups. I personally feel that the novelty of the presented data is very low, sadly. I think that conclusions, as well as the main findings of the study, are definitely known for a longer time and this paper does not add anything new to the field. I am aware that processing data, writing a paper, and sending it out takes time, but in the emerging fields the Authors should be quick as possible.

There are also some more limitations - the quick SOFA system is very basic. I recommend the Authors extract more data from the data set (since it is a retro study) and incorporate it into the paper - it might bright more novelty that might make your paper more "publishable".

Minors:

The limitations are fairly discussed, albeit, I miss data showing the real statistical power of the data analysis.

What is the reason for so large a number of patients with non-com[plete data sets? How it might impact the study results/conclusions?  

The data come from the period of March-October 2020 - why these frames? How might the virus mutations impact the results? 

Please revise carefully calculations, e.g. for COPD % - 44 patients mean 12.79%, not 12.9% as stated. Just minors but we should be very careful for typos. 

Best. 

Author Response

Replies to Reviewers’ comments – “A new early predictor of fatal outcome for COVID-19 in an Italian Emergency Department: the Modified Quick-SOFA” – Microorganisms-1665355 by Guarino et al.

Reviewer #2:

We wish to thank the Reviewer for her / his insightful comments. The following is a thorough point-by-point reply to each comment.

Major concerns

  1. My main concern about the paper is rather philosophical than scientific and is related to the novelty of the data shown there. I am not sure whether the paper is publishable in its present form - please take into account that similar studies have been done recently, with almost the same results (please refer to Frontiers in Medicine, 2021; JAMA, 2020; JAMA, 2021; Frontiers in Medicine, 2022; Blue Journal/ATS Scholar 2021; NEJM 2020 and much more...) on greater and more homogeneous groups. I personally feel that the novelty of the presented data is very low, sadly. I think that conclusions, as well as the main findings of the study, are definitely known for a longer time and this paper does not add anything new to the field. I am aware that processing data, writing a paper, and sending it out takes time, but in the emerging fields the Authors should be quick as possible.

Reply to comment 1: We take the Reviewer’s concern. We are aware of the literature on COVID-19 mortality prediction, but the purpose of this article was to propose a new, inexpensive and simple tool (i.e., Modified quickSOFA) that could be used in any medical setting, from the hospital - mainly in the Emergency Department - to outpatient clinics. This would be extremely important for the management of severe COVID-19 cases. Concerning the various articles cited by the Reviewer, they addressed different and more complicated scores, which are not suitable for the early assessment of patients in emergency settings. Taking sepsis as a clinical frame, as we all know, many scores have been proposed; however, MqSOFA has shown excellent mortality prediction ability, higher than qSOFA and non-inferior to NEWS and lactate assay (see Guarino et al. Infection. 2022. doi: 10.1007/s15010-022-01768-0). Finally, in this study, based on survival analysis according to the Kaplan-Meier method, the MqSOFA was non-inferior in predicting short-term (7 days) and mid-term (30 days) mortality risk in COVID-19 patients compared to the traditional NEWS-2 (7-day risk ratio: 5.27 vs. 4.31; 30-day risk ratio: 3.02 vs. 2.80).

  1. There are also some more limitations - the quick SOFA system is very basic. I recommend the Authors extract more data from the data set (since it is a retro study) and incorporate it into the paper - it might bright more novelty that might make your paper more "publishable".

Reply to comment 2: We thank The Reviewer for this comment. MqSOFA resulted superior in predicting the risk of short-term mortality compared to a more complex tool, such as NEWS-2. We elected to propose MqSOFA in the evaluation of COVID-19 patients because of its simplicity and, at the same time, effective evaluation of septic patients. As mentioned in the Reply to comment 1, this tool resulted superior to qSOFA and non-inferior to NEWS and lactate assay in sepsis although easier and non-invasive (for details, see Guarino et al. Infection. 2022. doi: 10.1007/s15010-022-01768-0). Since sepsis and COVID-19 share many features (see Dong X et al. Front Immunol. 2020. doi: 10.3389/fimmu.2020.598404) the aim of this paper was to propose the use of MqSOFA even in COVID-19 patients.

Minor concerns

  1. The limitations are fairly discussed, albeit, I miss data showing the real statistical power of the data analysis.

Reply to comment 1: We absolutely agree with the Reviewer. We added the data on the power analysis in the Results section (page 5, lines 183-184).

  1. What is the reason for so large a number of patients with non-complete data sets? How it might impact the study results/conclusions ?

Reply to comment 2: As this is a retrospective study, which does not directly involve patients, we could not recover missing data. Patients’ exclusion could appear as a potential bias; however, no prior selection has been done, thus we believe that their exclusion might not impact on the statistical analysis.

  1. The data come from the period of March-October 2020 - why these frames? How might the virus mutations impact the results?

Reply to comment 3: The study period was the 'first wave' of COVID-19, i.e. when respiratory complications and worse outcomes were more frequent due to the strong aggressiveness of the wild Wuhan strain and the incomplete knowledge of SARS-CoV-2 infection. In this emergency scenario, a simple tool such as MqSOFA was suitable for early identification of critical patients. We assume that most of the cases included in our study were infected with the wild-type SARS-CoV-2 strain, but we cannot specify the virus variants as no viral sequencing was being done at that time. In this regard, data involving large series of patients with identified variants are eagerly awaited.

  1. Please revise carefully calculations, e.g. for COPD % - 44 patients mean 12.79%, not 12.9% as stated. Just minors but we should be very careful for typos.

Reply to comment 4: All typos have been corrected and a new English proofread has been performed.

Reviewer 3 Report

This single-centre, retrospective study of 437 patients with SARS-CoV-2 infection aims to compare two different screening tools, a modified quick SOFA (MqSOFA) and the with NEWS-2 to predict in-hospital (IHM), 30-days mortality and recovery setting.

Comments.

  1. Some studies have verified the prognostic value of MqSOFA compared to lactate, NEWS and qSOFA in patients with sepsis. The present study indicates that the MqSOFA is non-inferior to NEWS-2 in predicting the outcome of patients with SARS-COV-2 infection. Although the current the study initially included 437 patients, one third of the patients were excluded due to inadequate data. Neither NEWS-2 nor MqSOFA were able to predict the intensity of care needed by the patient with SARS-CoV-2 infection. Could you explain why you only consider patients directly admitted to ICU (and not after 48-hours/7-days)?From a practical point of view, would it not be convenient to wait for a larger sample of patients to better assess the predicting value of the MqSOFA?.   
  2.   . Could you comment on the striking fact that the in-hospital mortality was extremely variable, ranging from 0 to 194 days. This wide variability is not frequent in other published series.                3. The SpO2 / FiO2 ratio has been widely used in all studies of patients with SARS-COV-2 infection. Would you mind commenting in more detail on the predictive limitations

Author Response

Replies to Reviewers’ comments – “A new early predictor of fatal outcome for COVID-19 in an Italian Emergency Department: the Modified Quick-SOFA” – Microorganisms-1665355 by Guarino et al.

Reviewer #3:

We wish to thank again the Reviewer for her / his insightful comments. The following is a thorough point-by-point reply to each comment.

  1. Some studies have verified the prognostic value of MqSOFA compared to lactate, NEWS and qSOFA in patients with sepsis. The present study indicates that the MqSOFA is non-inferior to NEWS-2 in predicting the outcome of patients with SARS-COV-2 infection. Although the current the study initially included 437 patients, one third of the patients were excluded due to inadequate data. Neither NEWS-2 nor MqSOFA were able to predict the intensity of care needed by the patient with SARS-CoV-2 infection. Could you explain why you only consider patients directly admitted to ICU (and not after 48-hours/7-days)? From a practical point of view, would it not be convenient to wait for a larger sample of patients to better assess the predicting value of the MqSOFA?

Reply to comment 1: We thank the Reviewer for her/his insightful comments. We considered only the direct admission to ICU in order to evaluate the ability of MqSOFA in detecting critical patients. A secondary patient transfer from a low- or medium-intensity ward to ICU was not considered in the objectives of this study. Instead, we chose to analyze the ability of involved tools in terms of mortality as the primary clinical endpoint (the recovery ward is considered a “surrogate endpoint”). We are aware of the limitation of this selection and therefore we are currently working to realize further prospective studies involving other patient subsets (e.g. those transferred from low- medium-intensity wards to ICU).

  1. Could you comment on the striking fact that the in-hospital mortality was extremely variable, ranging from 0 to 194 days. This wide variability is not frequent in other published series.

Reply to comment 2: The high mortality range is related to patients who were burdened by a large number of complications (i.e., respiratory, renal and bacterial superinfections). There was only one case in which the length of stay before death was 194 days. This case was anyhow included in the analysis.

  1. The SpO2 / FiO2 ratio has been widely used in all studies of patients with SARS-COV-2 infection. Would you mind commenting in more detail on the predictive limitations.

Reply to comment 3: We thank the Reviewer for this comment. A new paragraph on the SpO2/FiO2 ratio and its limitations has been added (see Discussion, pages 9-10, lines 291-296).

Round 2

Reviewer 2 Report

Dear Authors, 

Thank you for your very kind responses as well as for revising your paper. I truly appreciate it. 

I absolutely agree with you on many points, albeit, I still think that this paper lacks significance or rather novelty for clinicians. You utilize very basic parameters that have been shown in many different respiratory states as key predictors/scoring units that might predict outcome/mortality/morbidity. And at this point, you literally adapted them for Covid patients. I agree that this scoring model works and it is usable for this purpose. But I do not see any kind of novelty in your assumptions. 

I think that my work as the reviewer is done here - let Editor decide whether the level of novelty is fair enough to publish your paper. 

I appreciate your input in the covid area as well as your hard work as a scientist and clinician. As a trans-med scientist, I wish you all the best. 

Reviewer 3 Report

The answers to the questions raised have significantly improved the scope of the manuscript. Thank you very much